# Impact of Road Fencing on Ungulate–Vehicle Collisions and Hotspot Patterns

**Andrius Kučas [†] and Linas Balčiauskas ***

Nature Research Centre, Akademijos str. 2, LT-08412 Vilnius, Lithuania; andrius.kucas@gamtc.lt

* Correspondence: linas.balciauskas@gamtc.lt; Tel.: +370-68534141

† Present address: Territorial Development Unit, Growth and Innovation Directorate, Joint Research Centre, European Commission, Via Fermi 2749, I-21027 Ispra, Italy.

**Abstract:** The number of road traffic accidents decreased in Lithuania from 2002 to 2017, while the ungulate–vehicle collision (UVC) number increased and accounted for approximately 69% of all wildlife–vehicle collisions (WVC) in the country. Understanding the relationship between UVCs, traffic intensity, and implemented mitigation measures is important for the assessment of UVC mitigation measure efficiency. We assessed the effect of annual average daily traffic (AADT) and wildlife fencing on UVCs using regression analysis of changes in annual UVCs and UVC hotspots on different categories of roads. At the highest rates, annual UVC numbers and UVC hotspots increased on lower category (national and regional) roads, forming a denser network. Lower rates of UVC increase occurred on higher category (main) roads, forming sparser road networks and characterized by the highest AADT. Before 2011, both UVC occurrence and fenced road sections were most common on higher-category roads. However, as of 2011, the majority of UVCs occurred on lower-category roads where AADT and fencing had no impact on UVCs. We conclude that wildlife fencing on roads characterized by higher speed and traffic intensity may decrease UVC numbers and at the same time shifting UVC occurrence towards roads characterized by lower speed and traffic intensity. Wildlife fencing re-allocates wildlife movement pathways toward roads with insufficient or no mitigation measures.

**Keywords:** road safety; roadkill; clustering; growth rate; traffic intensity; mitigation measures



## 1. Introduction

Systems that record regular road traffic accidents, including those with animals, are continually evolving and becoming highly integrated [1]. In Lithuania, computerized wildlife–vehicle collision (WVC) reporting started in 2002, which integrates data stored at the Lithuanian Police Traffic Supervision Service with road data maintained by the Lithuanian Road Administration under the Ministry of Transport and Communication [2]. In the period 2002–2017, over 73,211 road traffic accident records were registered. The number of records decreased from 6090 in 2002 to 3192 in 2017 [3], constituting a 4.2% compound annual reduction (Table A1). In the period 2002–2017, total ungulate–vehicle collision (UVC) numbers in Lithuania have been constantly growing, constituting a 16.4% compound annual increase (Table A1).

WVCs present a serious problem and an increasing threat to traffic safety, socio-economics, animal welfare, and wildlife management and conservation in many countries [4–7]. The number of WVCs is steadily increasing in many countries [1,8–12] and in Lithuania [2,13].

Collisions with large mammals are a global and persuasive problem [14]. The rate of collision numbers has increased significantly over time, suggesting the growing importance of traffic in ungulate management. UVC is of particular importance from the perspective of drivers because of the large body size of the animals, resulting in a strong impact and consequences [15]. In contrast, small animals rarely cause traffic crashes, and the only

evidence of WVCs in such cases is the carcasses of the animals along roads, which can be a proxy for the extent of such events [1,8].

Knowledge of sites where UVCs occur more frequently is important for the effective application of mitigation measures [6,16–21] and identifying significant locations where animal pathways intersect with roads. Collision risks may be associated with linear landscape features that funnel animals alongside or across the road and with artificial infrastructure [4,22].

A variety of UVC mitigation measures may be implemented to modify the behavior of drivers and/or animals in order to reduce the number of UVCs. These include measures to physically block animal movement on roads. Wildlife fencing along roads in conjunction with the construction of wildlife passages has been widely accepted as an effective way to minimize collisions with animals [23]. However, there is limited information on the effectiveness of mitigation [24,25].

We analyzed UVCs in Lithuania on different categories of roads in the period 2002–2017, aiming to (i) map UVCs for each year, (ii) identify yearly UVC hotspots (short significant road segments where UVCs occurs more frequently than expected), and (iii) analyze UVCs and UVC hotspot relationships with annual average daily traffic (AADT) and the length of fenced road sections accounting for yearly changes.

We tested the following two hypotheses:

1. The occurrence of UVCs (and consequently UVC hotspots) directly depends on transport intensity (that is, UVC numbers will be bigger on the main roads, which are characterized by higher levels of speed and traffic).
2. Wildlife fences are sufficient measures for UVC prevention (that is, no UVCs or UVC hotspots will be recorded within the fenced road sections).

## 2. Materials and Methods

We collected and mapped UVC data in our study area. Using this data we identified UVC hotspots that allowed us to identify UVC spatial locations on different categories of roads. Compound annual growth rates used to identify long term change patterns of UVCs, UVC hotspots, AADT, and fence length on different categories of roads. Finally we assessed the effect of AADT and wildlife fencing on UVCs using regression analysis of changes in annual UVCs and UVC hotspots on different categories of roads.

### 2.1. Study Area

Our study area covered the entire territory of Lithuania (Figure 1). The country is located in northern Europe and borders the Baltic Sea. The flat area (with the highest point of ~294 m above sea level) of the country extends to 65,286 square kilometers. Lithuania's climate is transitional between maritime and continental regions. The average air temperatures are –4.9 °C in January and 17.2 °C in July. Annual rainfall average is from 570 mm to 902 mm, depending on the location [26].

The country is located in a mixed-forest zone; 33% of the surface is occupied by arable land and permanent crops, 27% by semi-natural vegetation, 33% by forested land, 3% by artificial areas, and 4% by water bodies and other land [27]. The country is inhabited by 68 species of mammals, including eight species of ungulates [28].

In 2017 there were 21,244 km of state-owned roads of national significance (excluding roads in cities). While sources differ in terms of their exact nomenclature, the basic hierarchy comprises freeways, arterials, collectors, and local roads [29]. In Lithuania, main roads or highways can be considered as freeways or motorways (total length 1865 km with AADT 3000–20,000 cars per day), national roads as arterials and collectors/distributors (5006 km, 500–3000 cars per day), and regional roads as local roads (14,600 km, up to 500 cars per day) [30]. The AADT values for different road categories are provided in Table A1.

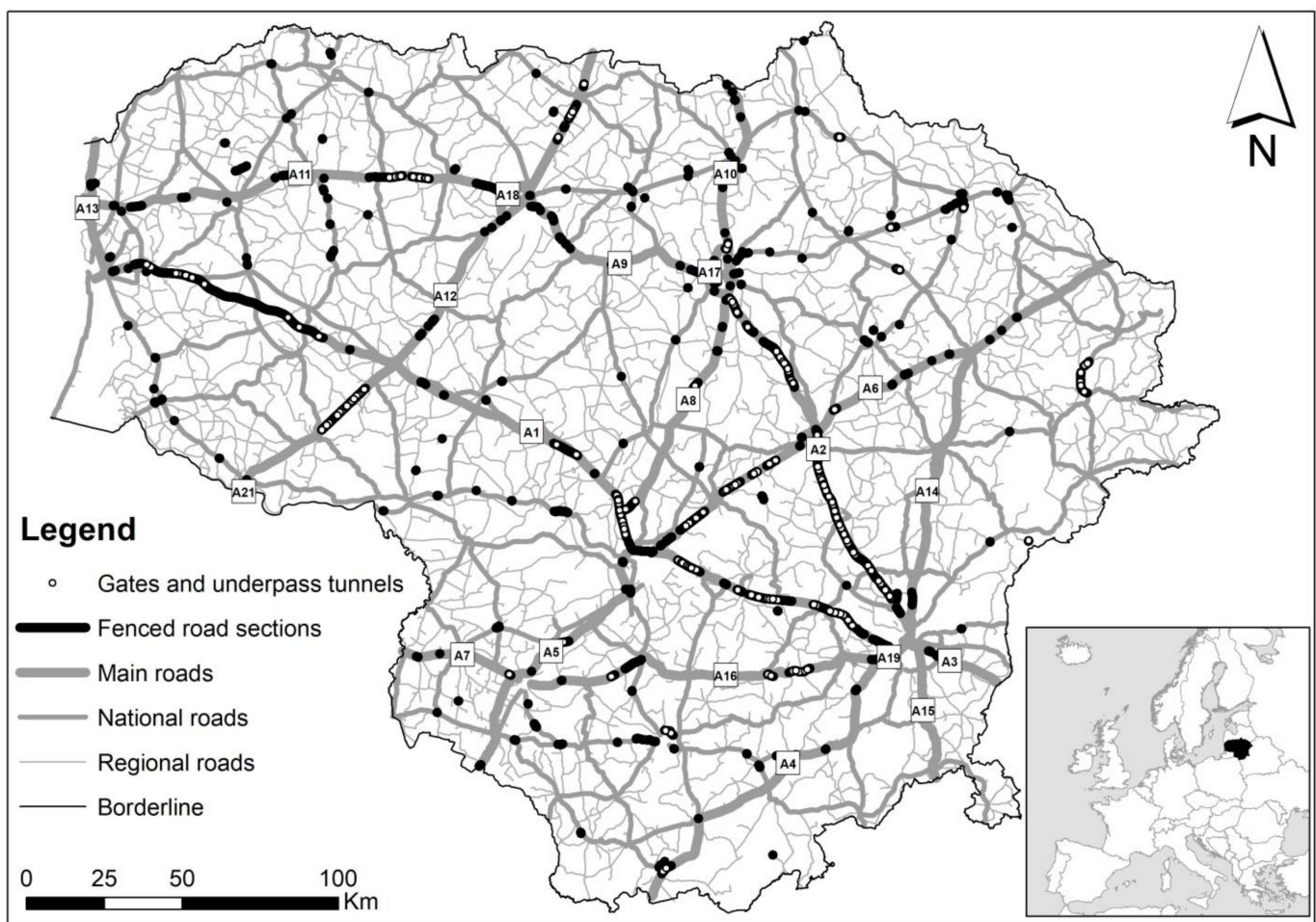

**Figure 1.** Study area, roads by categories, and fenced road sections in conjunction with additional mitigation measures such as underground passes, gates, etc., in 2002–2017. Labels show the unique identification numbers of main roads/highways.

The width of traffic lane ranges from 2.0 m to 3.75 m. The width of road lanes (traffic lanes including shoulders and safety rails) depends on five different road categories and ranges from 18 m (1–2 lanes) to 39 m (4–6 lanes separated by grass line).

Wildlife fences are the most common UVC mitigation measure in Lithuania. In addition to wildlife fences, other UVC mitigation measures (underground passages, tunnels, gates and jump outs) have been implemented [3].

In 2017 only 3.78% of all roads were fenced in Lithuania. There were 1088 (total length 803.5 km) segments of wildlife fences, 680 of which (743.8 km) were implemented on main roads, 256 (48.6 km) on national regional roads, and 152 (11.1 km) on regional roads (Figure 1).

### 2.2. Ungulate–Vehicle Collision Data

UVCs occur in approximately 47% of the entire Lithuanian road network (not including cities). According to data from the Lithuanian Police Traffic Supervision Service and Nature Research Centre, a total of 21,847 WVCs (15,006 UVCs) were recorded over the period of 2002–2017 in Lithuania (Table A1). These numbers may not account for all accidents, as the reporting of accidents with all WVCs to the authorities is not mandatory in Lithuania. However UVCs are reported in most cases, since reporting is mandatory in cases where animals and/or people involved in the accident are killed and/or injured, or vehicles and/or road infrastructure are damaged.

We selected 13,762 UVC reports that had coordinates and involved six species of large mammals (Figure 2, Tables A1 and A2): 1340 moose (*Alces alces*), 248 red deer (*Cervus elaphus*), 10,741 roe deer (*Capreolus capreolus*), 1416 wild boar (*Sus scrofa*), 11 fallow deer

(*Dama dama*), and 6 European bison (*Bison bonasus*). These large animals caused the most serious accidents and in the vast majority of cases were registered by the police [31]. So far there have been no UVCs reported with European mouflon (*Ovis aries*) or sika deer (*Cervus nippon*).

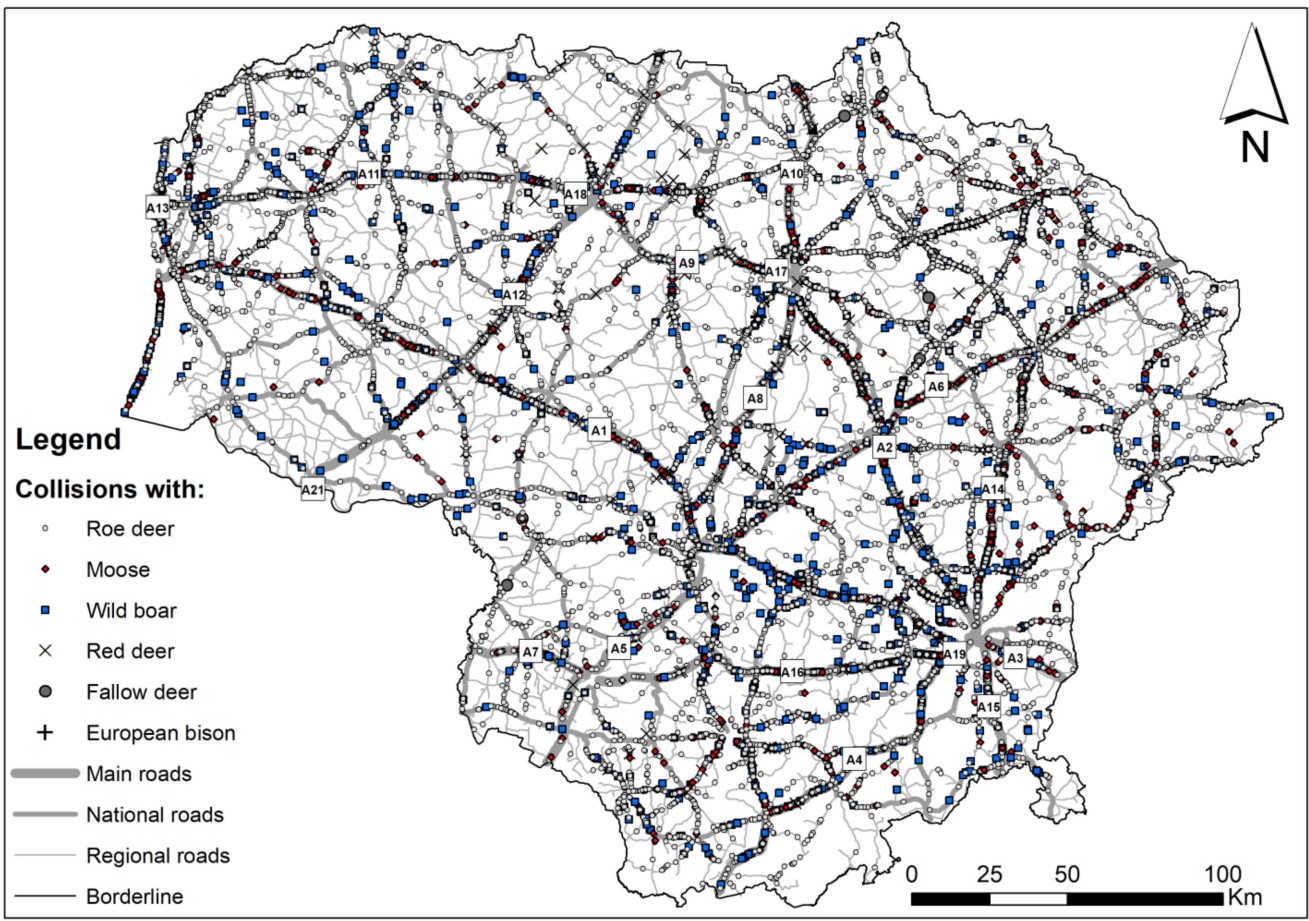

**Figure 2.** Locations of the ungulate–vehicle collisions (UVCs) in Lithuania, 2002–2017.

### 2.3. Identification of Hotspots

The literature reports many different spatial techniques to identify relatively short road segments where road accidents or hazards occur more frequently than expected [4,16,32–38]. We performed UVC data clustering using the KDE+ tool [16,39], which analyses UVCs, represented as point features, located along the roads that are represented as line features. The KDE + tool identifies short significant road sections (so-called "clusters") where accidents occur more frequently than expected. The tool also assigns strength values (measured from 0 to 1) to the clusters, showing the highest probability of a crash from the driver's point of view [16]. We created UVC clusters (hereinafter "UVC hotspots") using the recommended parameters [16] by following road network properties (KDE+ bandwidth, 150 m; Monte Carlo simulations, 800; and minimal cluster strength, 0.2).

### 2.4. Compound Annual Growth Rates

We analyzed the UVC, UVC hotspot, AADT, and fencing compound annual growth rate (CAGR) on the different categories of roads. CAGR is defined as [40]

$$\mathrm{CAGR}\,(t_0, t_n) = \left(\frac{V(t_n)}{V(t_0)}\right)^{\frac{1}{t_n - t_0}} - 1, \tag{1}$$

where $V(t_0)$ is the initial value of AADT, UVC, and UVC hotspot numbers and fence lengths, $V(t_n)$ is the end value of the same parameters, and $t_n - t_0$ is the number of years. We used CAGR to smooth variable returns so that they could more easily be used for evaluation of long-term UVC, UVC hotspot, AADT, and fence length changes that occur in different road categories.

### 2.5. Multiple Linear Regression Analysis

We assessed the effect of wildlife fencing on UVC in Lithuania from 2002 to 2017 by performing multiple linear regression (MLR) analysis [41]. The UVC and UVC hotspots were the dependent variables. We checked the UVCs and UVC hotspots against AADT and fence lengths on main, national, and regional roads.

We checked how UVCs and UVC hotspots on one category of roads depend on UVCs and UVC hotspots on the other two categories of roads. Then we checked how UVCs on one category of roads depend on AADT on all categories of roads. Last we checked how UVCs and UVC hotspots on one category of roads depend on fence lengths on all categories of roads. We ran models separately for every category of roads.

In all MLR, regression coefficients b were treated as indicators of strength of the effect of each individual independent variable to the dependent variable.

Finally, we used a unified modelling language (UML) collaboration diagram [42] to describe the results of the multiple regression analyses.

## 3. Results

We identified UVC, UVC hotspot, AADT, and fencing change patterns on different categories of roads. To test our hypotheses we used regression analyses. We identified relationships between UVCs and UVC hotspots on different categories of roads. In addition, we identified UVC and UVC hotspot relationships with fencing and AADT on different categories of roads.

### 3.1. Roadkill Hotspots

We identified UVC hotspots every year from 2002 to 2017. In total, we found 691 unique UVC hotspot locations (Figure 3) for all categories of roads (261, 373, and 57 hotspots on main, national, and regional roads, respectively), with the hotspots having an average length of 133 m (Table A3) and a length range of 100–471 m. The range of strength (KDE+) of UVC hotspots on main, national, and regional roads was 0.27–0.50, 0.39–0.50, and 0.33–0.48, respectively (Table A3). The UVC hotspots resulting from the analysis are also accessible online as a web map service [43].

### 3.2. Patterns of UVC, AADT, and Fencing Changes

On average, 938 UVCs were recorded each year for the period 2002–2017 (Table A1). UVCs including roe deer, red deer, moose, fallow deer, bison, and wild boar accounted for approximately 69% of WVCs. Roe deer alone accounted for approximately 54% of all UVCs.

The regional road network was the largest and densest, while the national road network was denser than the main road network but sparser then the regional road network (Figure 1). The highest AADT was found on the main roads and the lowest on the regional roads (Table A3). The largest share of UVCs and UVC hotspots occurred on national roads (Figure 4). Decreasing AADT and increasing UVC hotspots on regional roads suggest that AADT does not impact UVCs on regional roads characterized by lower speed and traffic intensity.

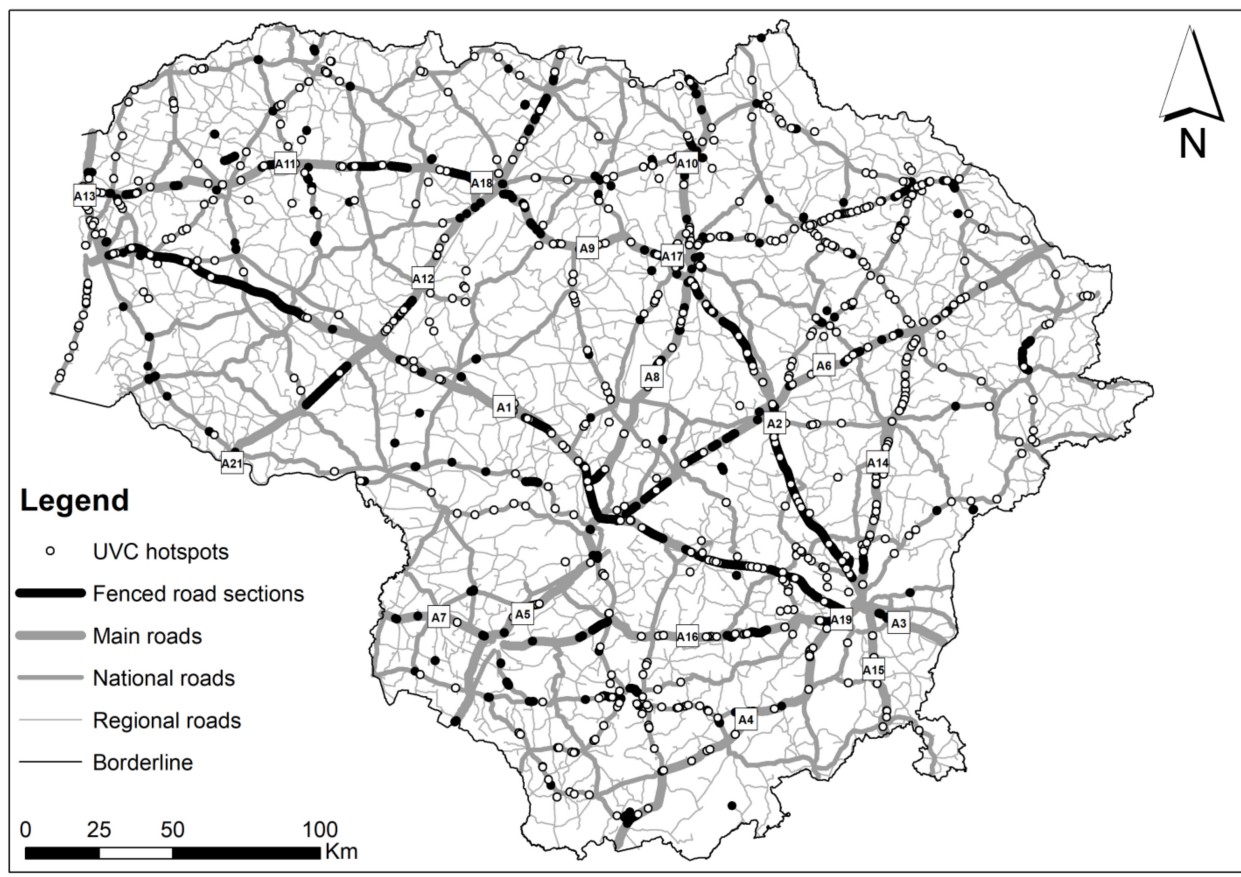

**Figure 3.** UVC hotspots and fenced road sections in the 2002–2017 period.

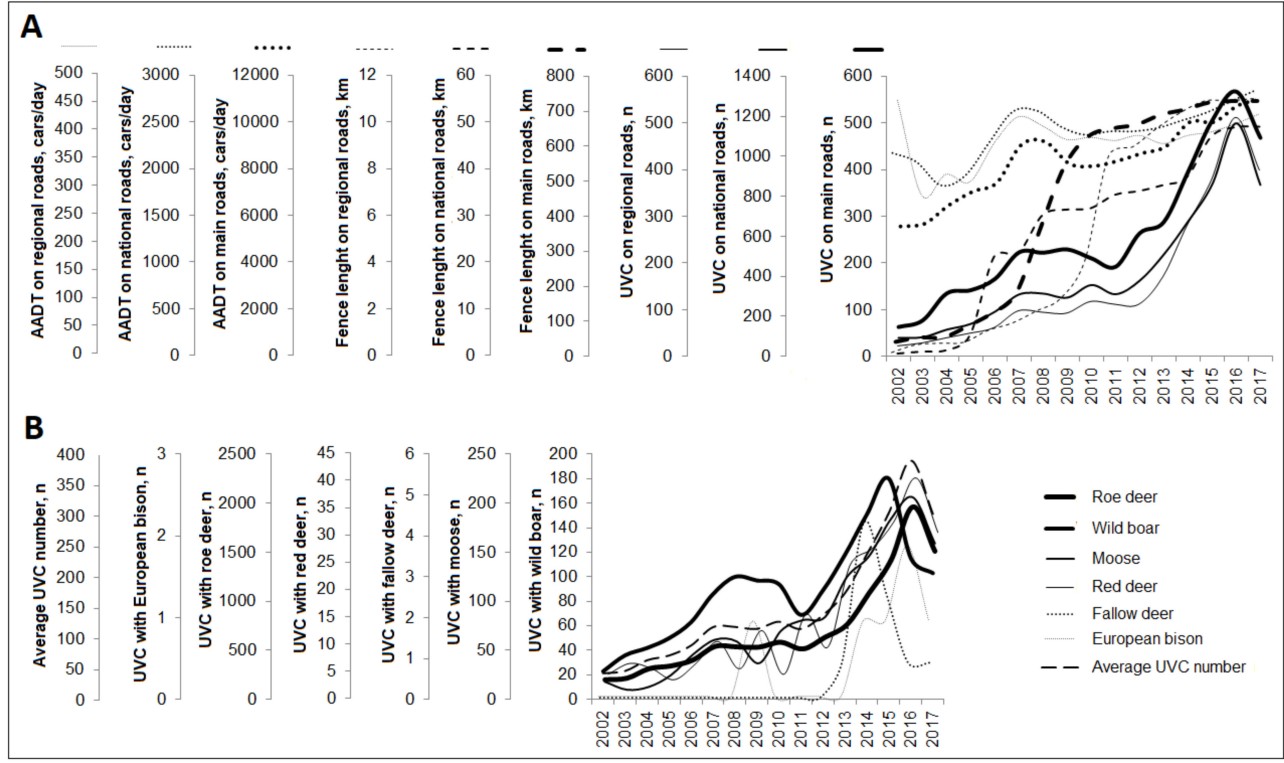

**Figure 4.** Change patterns of annual average daily traffic (AADT), including all types of vehicles, total UVCs (**A**, Table A1), fencing (**A**, Table A4), and species-specific UVCs (**B**, Table A2) in 2002–2017.

CAGR analysis revealed that AADT increased on main and national roads, while it decreased on regional roads (Figure 5, Table A1). UVC hotspot number, length, and average strength increased the most on regional roads (Table A3). The UVC hotspot average strength decreased on the main and regional roads (Table A3). The length of the new fences decreased the most on the main roads and increased on regional roads only (Table A4). The total length of fences increased the most on national roads (Figure 5). However, the share of new fences was decreased on the national and main roads (Figure 5, Table A1).

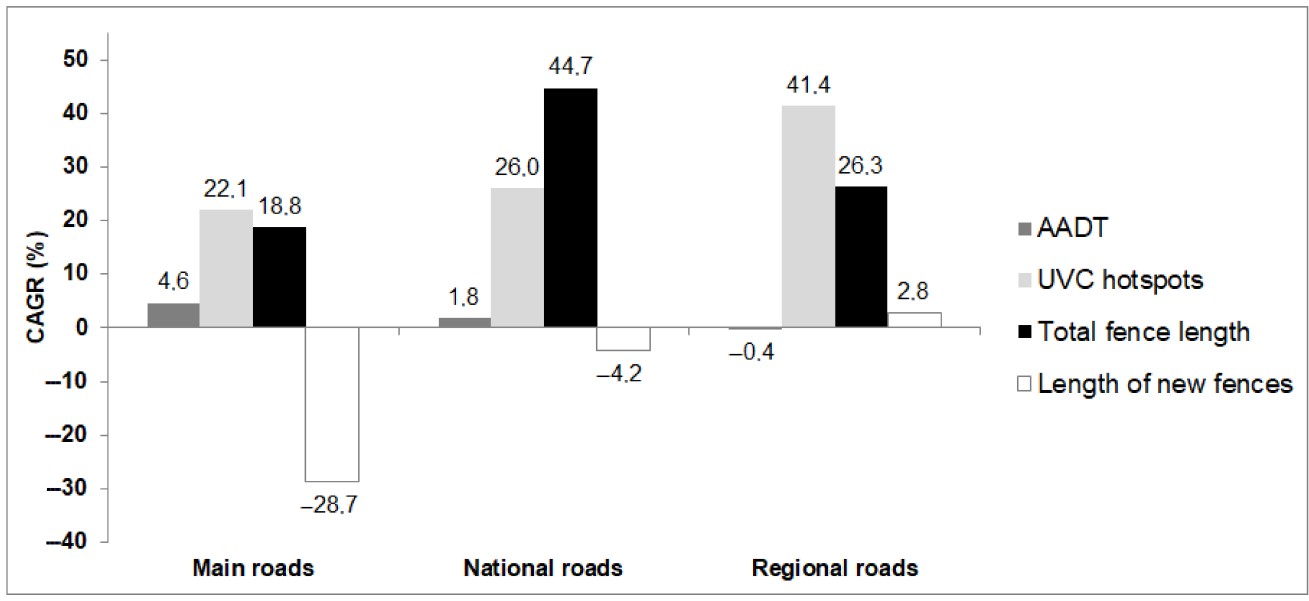

**Figure 5.** Compound annual growth rate (CAGR) of AADT (Table A1), UVC hotspot (Table A3), and fence length (Table A4) distributions within the different types of road networks in 2002–2017.

CAGR analysis suggested that wildlife fencing on roads characterized by higher speed and higher traffic intensity (sparser network) may shift collision occurrence towards roads (denser road network) characterized by lower speed and lower traffic intensity. To confirm or reject this assumption, we performed an additional MLR analysis.

### 3.3. Factors Influencing Roadkills

The patterns of relationship of UVCs and UVC hotspots on different road categories are shown in Table 1. We checked how changes of UVC and UVC hotspots on regional roads were influenced by dynamics of UVC and UVC hotspots on national and main roads, repeating the model for all road categories.

**Table 1.** Hypotheses that there are no relationships between UVC and UVC hotspot patterns on different categories of roads declined with * $p < 0.10$, ** $p < 0.05$, *** $p < 0.01$, **** $p < 0.001$. NS: not significant. Response variables are UVCs and UVC hotspots on regional, national, and main roads; $b_{UVC}$ show the degree of change in the response variable for every 1-unit of change in the predictor variable.

| Target | | Intercept ± SE | $b_{UVC}$ ± SE | | F | R² |
|---|---|---|---|---|---|---|
| UVC | Regional | −34.17 ± 13.08 ** | National | Main | 477.29 **** | 0.99 |
| | | | 0.54 ± 0.09 **** | −0.15 ± 0.20 * | | |
| | National | 18.62 ± 24.69 NS | Regional | Main | 864.69 **** | 0.99 |
| | | | 1.33 ± 0.23 **** | 0.77 ± 0.23 *** | | |
| | Main | 40.42 ± 19.48 **** | Regional | National | 246.63 **** | 0.97 |
| | | | −0.23 ± 0.38 NS | 0.61 ± 0.18 *** | | |

**Table 1.** *Cont.*

| Target | | Intercept $\pm$ SE | $b_{UVC} \pm$ SE | | F | $R^2$ |
|---|---|---|---|---|---|---|
| | | | $b_{UVC}$ hotspots $\pm$ SE | | | |
| UVC hotspots | Regional | $-1.47 \pm 0.89$ NS | National | Main | 34.05 **** | 0.84 |
| | | | $0.07 \pm 0.08$ NS | $-0.21 \pm 0.14$ NS | | |
| | National | $-0.94 \pm 3.13$ NS | Regional | Main | 79.06 **** | 0.92 |
| | | | $6.69 \pm 0.89$ NS | $1.34 \pm 0.30$ **** | | |
| | Main | $3.20 \pm 1.59$ ** | Regional | National | 90.20 **** | 0.93 |
| | | | $0.74 \pm 0.47$ NS | $0.45 \pm 0.10$ **** | | |

The relationship patterns of UVC and UVC hotspots with AADT and fencing length on different categories of roads are shown in Table 2. We checked if UVC number on the regional, national, and main roads was related to AADT of all these road types. MLR with UVC hotspot number regressed to AADT were all not significant and therefore are not presented in Table 2. Then we checked if UVC numbers and UVC hotspot numbers were related to the length of fencing on all road categories.

**Table 2.** Hypotheses that there are no relationship between UVC and UVC hotspots with AADT and fencing length patterns on different categories of roads declined with * $p < 0.10$, ** $p < 0.05$, *** $p < 0.01$, **** $p < 0.001$. NS: not significant. Response variables are UVCs and UVC hotspots on regional, national, and main roads; $b_{AADT}$ and $b_{length\ of\ fences}$ show the degree of change in the response variable for every 1-unit of change in the predictor variable.

| Target | | Intercept $\pm$ SE | Regional | National | Main | F | $R^2$ |
|---|---|---|---|---|---|---|---|
| | | | $b_{AADT} \pm$ SE | | | | |
| UVC | Regional | $-405.17 \pm 230.81$ * | $0.03 \pm 0.80$ NS | $-0.08 \pm 0.03$ NS | $0.09 \pm 0.03$ ** | 10.49 ** | 0.72 |
| | National | $-828.65 \pm 432.42$ * | $0.13 \pm 1.50$ NS | $-0.16 \pm 0.44$ NS | $0.19 \pm 0.06$ *** | 14.13 **** | 0.78 |
| | Main | $-320.62 \pm 172.45$ NS | $0.03 \pm 0.60$ NS | $-0.12 \pm 0.18$ NS | $0.10 \pm 0.02$ *** | 22.30 **** | 0.84 |
| | | | $b_{lenght\ of\ fences} \pm$ SE | | | | |
| UVC | Regional | $12.25 \pm 38.84$ NS | $29.09 \pm 10.83$ ** | $8.05 \pm 3.35$ ** | $-0.49 \pm 0.24$ * | 14.80 ** | 0.79 |
| | National | $86.55 \pm 72.99$ NS | $53.82 \pm 20.35$ ** | $17.35 \pm 6.29$ ** | $-0.90 \pm 0.45$ * | 19.39 **** | 0.83 |
| | Main | $83.02 \pm 32.34$ ** | $23.07 \pm 9.01$ ** | $9.15 \pm 2.79$ *** | $-0.43 \pm 0.20$ ** | 23.47 **** | 0.85 |
| UVC hotspots | Regional | $-1.03 \pm 1.61$ NS | $1.45 \pm 0.45$ *** | $0.25 \pm 0.14$ * | $-0.02 \pm 0.01$ ** | 11.65 **** | 0.74 |
| | National | $1.64 \pm 8.58$ NS | $4.03 \pm 2.39$ NS | $1.74 \pm 0.74$ ** | $-0.10 \pm 0.05$ * | 7.96 ** | 0.67 |
| | Main | $1.46 \pm 4.76$ NS | $3.30 \pm 1.33$ ** | $0.97 \pm 0.41$ ** | $-0.06 \pm 0.03$ ** | 22.30 ** | 0.73 |

In order to simplify the interpretation of the MLR results (Tables 1 and 2), we used the UML collaboration diagram (Figure 6), which showed only significant relationships. The vertical swim lanes in Figure 6 represent the different road categories. The dependent and independent variables were represented as rectangles. The first sign on the right (+ or −) adjacent to the line indicates changes within the source variables, while the second sign on the left (+ or −) indicates changes in the target variables (effect). Arrows show significant source–target relationships (Tables 1 and 2). For instance, an increase (+) in fence length on main roads was significantly related to the decrease (−) of UVC and UVC hotspots on main roads (Table 2). The relationship lines are absent in cases where no significant relationship was found between variables (for instance, between AADT and UVC on national and regional roads).

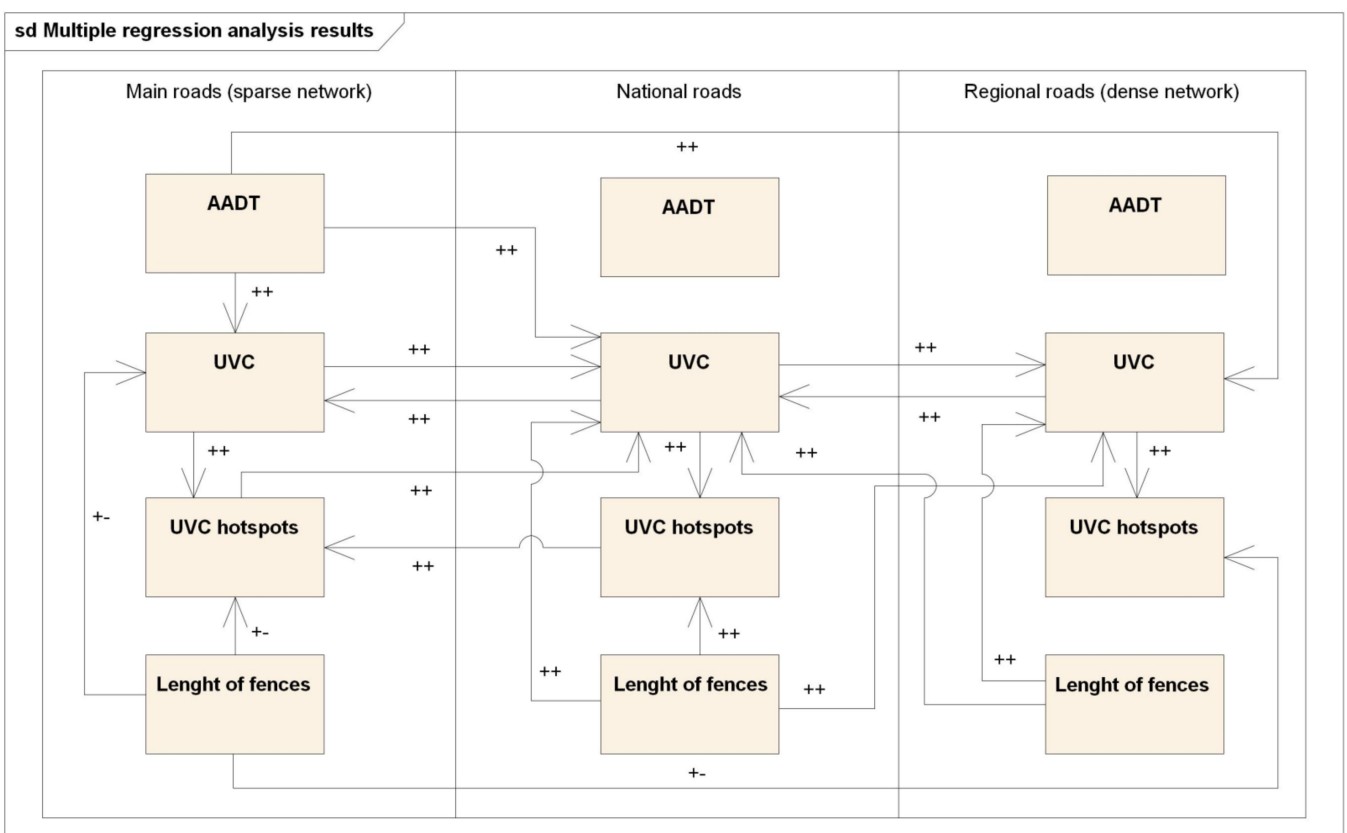

**Figure 6.** Unified modelling language (UML) collaboration diagram that shows MLR analysis results.

MLR analysis results show that while AADT increased on main roads, the UVC numbers increased on all lower category roads (Table 2, Figure 6). These relationships show that AADT had no effect on the UVC on national and regional roads (no relationship lines in Figure 6). However, an increase in AADT on higher-category roads may shift collision occurrence towards roads characterized by lower speed and lower traffic intensity.

As UVC increased on main roads, UVC became more frequent on both national and then regional roads. However, the increase in UVC on main roads did not directly impact UVC growth on regional roads (Table 1, Figure 6). As UVC increased on regional roads, UVC became more frequent on both national and main roads. However, the increase in UVC on regional roads did not directly impact UVC growth on the main roads (Table 1, Figure 6). These relationships show that changes in UVC on higher category roads have a direct relationship with changes in UVC on lower-category roads and vice versa.

An increase in UVCs increased UVC hotspots on the main roads. The same UVC and UVC hotspot relationships were also observed on national and regional roads. An increase in UVC hotspots on main roads increased UVCs on national roads. However, an increase in UVC hotspots on national roads increased UVC hotspots on main roads (Table 1, Figure 6). These relationships showed that changes in UVCs have a direct relationship with changes in UVC hotspots on all types of roads. However, an increase in UVC hotspots on lower-category roads might increase UVC hotspots on higher-category roads.

An increase in the length of wildlife fencing on the main roads diminished the number of UVCs and UVC hotspots (Table 2, Figure 6). This relationship showed that fences are effective mitigation measures for main roads.

The length of wildlife fencing on national and regional roads increased the UVC on national and regional roads (Table 2, Figure 6). These relationships showed that fencing had no effect on the UVCs on lower-category roads.

As the length of the fences increased, so did the UVC hotspots on national roads. As the length of the fences increased, so did the UVCs on regional roads. These relationships

showed that fencing was not as effective as expected on lower-category national and regional roads (Table 2, Figure 6).

## 4. Discussion

First we discuss relationship between UVCs, AADT, and fencing. We partly confirm our hypotheses and explain changes in roadkill and fencing patterns, and evaluate wildlife fences on different categories of roads. We also provide recommendations on how to improve UVC mitigation measures on different categories of roads.

### 4.1. Relationship between UVC, AADT, and Fencing

MLR results confirm both tested hypotheses for the main roads. We acknowledge that the pattern of UVC numbers on the highest-category (main) roads is explained by changes in AADT. Moreover, the growth of AADT on main roads might increase the UVCs on national and regional roads (Table 2, Figure 6). Correlations between the number of UVCs and AADT on lower category (national and regional) roads were insignificant, and this is in agreement with [12,44].

We also found that the growing length of wildlife fencing on the main roads effectively diminished the number of UVCs and UVC hotspots (see Table 2 and Figure 6, links between UVCs, UVC hotspots, and fence length on main roads). However, main road fencing redirected wildlife towards lower category roads and dispelled UVC hotspots on regional roads (see Table 2 and Figure 6, link between fences on main roads and UVC hotspots on regional roads).

### 4.2. Changes in Roadkill and Fencing Patterns

A pattern is regular in the UVC, UVC hotspot, and fencing locations on different categories of roads over time. During 2002–2017 in Lithuania the number of UVCs increased in all categories of roads. At the beginning of the period, both UVCs and fenced road sections were most common on main roads characterized by higher speed and traffic intensity. The same pattern was observed in Spain [45]. However, as of 2011, the numbers of both UVC and fenced road sections started to grow on roads characterized by lower speed and traffic intensity (Figure 4A). Similar to other countries [46,47] this UVC increase can be an effect of increasing wildlife populations in Lithuania [13].

Changes in UVC patterns may be related to blocked wildlife pathways due to frequent fencing on the main roads. While the number of UVCs on fenced road sections has been reduced, it is growing on non-fenced road sections and adjacent roads (Figure 6, Table 2). It might require time for wildlife to rebuild new pathways (e.g., Figure 4A, fencing before 2011, UVC 2008–2012 period). Scattering of the new UVCs is the reason why UVCs do not form hotspots (Figure 6). After new wildlife paths are established, UVC hotspots start to occur in new locations on lower-category roads.

However, we assume that the formation of hotspots shows stability in ungulate pathways. We found that annually only 16% of all UVCs occurred within hotspots, which suggests that a large part of wildlife pathways in the country are scattered and not permanent. We partly confirmed the results of [7,48–50] that UVCs are not spatially random, since 84% of UVCs (2002–2017) were not located in the hotspots.

From the above we conclude that in the short-term, wildlife fencing can decrease UVC numbers on main roads, but as a result of altered wildlife pathways, UVC locations shift towards the denser lower category road network. At the same time, limited movements may reduce the importance of adjacent habitats for wildlife and may amplify the importance of more distant habitat patches [51].

### 4.3. Evaluation of Wildlife Fences

WVCs are a cause of serious concern for road planners and biologists in terms of traffic safety, species conservation, and animal welfare [4]. Collisions numbers can be significant to species conservation, wildlife management, and traffic safety, thus creating ethical, social,

economic, and even political tensions. Putting in place mitigation measures is challenging, due to the lack of knowledge on the local spatial–temporal patterns of wildlife dynamics, including population, behavior, pathways, and habitat suitability [15,52,53].

Wildlife fences in conjunction with underpasses, gates, and jump outs are the most common WVC mitigation measures in Lithuania [54]. So far, no overpasses or advanced dynamic wildlife warning systems have been deployed, and the effectiveness of mitigation measures is still discussed in the country. The reason is that the location of fences is fixed, while the behavior of different species constantly changes [20,55]. A better understanding of the spatial distribution of WVC [4,20] requires consideration of many factors. They include, but are not limited to, understanding of wildlife movements and behaviors, localization of wildlife corridors [51,56–58], knowledge of population density [59], population dynamics and habitat properties. In line with other authors [15,52,53,60] we also confirm that placing mitigation measures is challenging, because of the lack of knowledge on the local spatiotemporal patterns of wildlife dynamics, including population, behavior, pathways and habitat suitability [13,20].

Exponential growth of the length of installed wildlife fences occurred in 2008–2011, and since 2011 fencing intensity considerably decreased (Figure 4A). As the outcome, an increase in UVCs was observed after 2012, when ungulates adapted to existing fencing. The longest sections of the wildlife fences on the main roads of Lithuania were installed in 2004–2010, on national roads in 2005–2008, and on regional roads in 2009–2012 (Figure 4A, Table A4). Thus, safety measures targeting roads with lower traffic intensity and speed were introduced at the end of the analyzed period. UVC and UVC hotspot occurrences constantly increased during 2004–2016. In 2017, the longest fenced road sections were on main roads, while regional roads had minimal fencing.

The highest rate of UVC on national roads (50.7%, Table A1) conforms to the fact that only 6.1% of all wildlife fences were installed there (Table A4). We concluded that the increase in the AADT and length of wildlife fences on the main roads shifted UVCs, first towards national roads and later towards regional road networks (Figure 6). The highest number of UVCs (especially with roe deer) was on national roads. In contrast to another study [61,62], fewer accidents were caused by ungulates on the main roads where the traffic volume was greater and speed was higher, as 93% of wildlife fences were installed along them (Table A4). Fencing is effective and may reduce roadkill rates [4,24], but on highest-category roads only (Table 2, Figure 6). Thus, the efficiency of fences decreases in the lower categories of the road network (Table 2, Figure 6 national and regional roads).

We found only a few new fences in the locations where UVC hotspots occurred in the previous year. This may be due to the fact that wildlife fencing in Lithuania is not based on WVC data [2] and is organized according to the strictly defined road infrastructure reconstruction programs. Such programs address the road safety standards for main roads/highways, rather than adequately responding to constantly changing UVC situations on lower-category roads.

Short wildlife fences may not sufficiently reduce the risk of accidents [63–66]; however, they are economically more affordable. Long fences are less economically efficient, but may perform better [18,63,66] on roads with the highest traffic intensity [67]. Building longer fences because of traffic safety reasons may be unduly costly [63], especially on dense road networks. From a wildlife perspective, longer fences cause landscape and habitat fragmentation and isolation of populations [66,68]. Therefore, they require additional measures to enable safe road crossings for wildlife.

Fencing not only prevents ungulates from crossing roads, but also directs them to the passage infrastructure. This might force animals to avoid roads with higher traffic and speed intensity. Mitigation measures may redirect animal pathways towards more attractive and distant habitat patches; however, they inevitably contribute to increasing UVC numbers on the lower category roads characterized by lower speed and traffic intensity (Table 2, Figure 6).

As a standard, all highways characterized with the highest traffic and speed intensity have to be fenced due to traffic safety reasons. Consequently, UVC rates grew on the lower category roads where no UVC mitigation measures were deployed (Table 2, Figure 6). Continuing building fences on unprotected main road sections without proper planning [69] can shift the problems to unfenced national and regional road sections. In addition, it may disconnect important habitats and may become the reason for serious ecological problems such as the extinction of local populations [70] and discontinuity of important ecological networks [51].

In line with [23,24,65] we confirm that wildlife fences are an effective long-term UVC mitigation measure on highways. However, this measure can only be effective if fences are planned in a timely manner [68,69,71], carefully inserted in the landscape [72], properly maintained, and in conjunction with other permanent UVC mitigation measures such as underpasses, overpasses, and driver warning systems. If not, habitat isolation may be amplified, and costs of construction and maintenance may be too high without any positive effect on the drivers and wildlife, especially on lower-category roads [65].

Modifying the natural behavior of ungulates is almost impossible [10]. Consequently mitigation should focus on the modification of human behavior and changing drivers' attitudes [73], introducing novel car safety systems [74], and improving road engineering [10,75].

There were no significant relationships between fences and UVC hotspots on regional roads; moreover, increases of fencing length resulted in increases of UVC hotspot numbers on national roads (Table 2, Figure 6). Therefore, the effectiveness of wildlife fences on national and regional roads is limited. We recommend less restrictive types of mitigation measures (e.g., advanced dynamic wildlife warning systems not preventing wildlife crossings) that should be applied for short significant road sections (hotspots) on lower (national and regional) category roads.

## 5. Conclusions

Analysis of the relationships between UVCs, UVC hotspots, fencing, and AADT in Lithuania showed the following:

1.  Wildlife fences are an effective mitigation measure for the main roads characterized by the highest traffic intensity. Fencing is not effective on lower-category roads where traffic intensity has a less significant impact on UVCs.
2.  Increased amounts of wildlife fencing may reduce the number of UVCs on the main roads and shift UVCs toward national and regional roads, characterized by lower speed and traffic intensity (denser road network).
3.  We recommend that efforts to reduce wildlife collision occurrence on lower-category roads should focus on driver attitudes and road conditions, rather than animal movement and behavior.

**Author Contributions:** Conceptualization, A.K. and L.B.; methodology, A.K.; software, A.K.; validation, A.K.; formal analysis, A.K.; investigation, A.K. and L.B.; resources, L.B.; data curation, L.B.; writing—original draft preparation, A.K.; writing—review and editing, L.B.; visualization, A.K.; supervision, L.B.; project administration, L.B.; funding acquisition, L.B. Both authors have read and agreed to the published version of the manuscript.

**Funding:** This research received no external funding.

**Data Availability Statement:** Spatial data for this article are published as a web map journal https://www.arcgis.com/apps/MapJournal/index.html?appid=d81195212a4b4bcc9c5aab34a0037609 (accessed on 1 March 2021) and can also be found online at http://dx.doi.org/10.17632/8gdpz5845x.4 (accessed on 1 March 2021).

**Acknowledgments:** The authors would like to sincerely thank to K. Tóth, for comments and anonymous reviewers for their suggestions and contributions to this manuscript.

**Conflicts of Interest:** The authors declare no conflict of interest.

## Appendix A

**Table A1.** The total number of road accidents, traffic intensity (AADT), and ungulate–vehicle collisions (UVCs) on different categories of roads.

| Year | Total Road Accidents | AADT | | | Total UVC | UVC [1] | | |
|------|---------------------|------|----------|----------|-----------|------|----------|----------|
| | | Main | National | Regional | | Main | National | Regional |
| 2002 | 6090 | 5610 | 2178 | 451 | 197 | 63 | 108 | 27 |
| 2003 | 5963 | 5729 | 2082 | 282 | 221 | 77 | 111 | 33 |
| 2004 | 6372 | 6519 | 1833 | 319 | 328 | 135 | 149 | 44 |
| 2005 | 6771 | 7107 | 1932 | 306 | 376 | 142 | 178 | 54 |
| 2006 | 6658 | 7488 | 2288 | 375 | 470 | 166 | 239 | 66 |
| 2007 | 6448 | 9100 | 2624 | 422 | 700 | 223 | 325 | 102 |
| 2008 | 4795 | 9240 | 2630 | 406 | 689 | 222 | 329 | 99 |
| 2009 | 3827 | 8293 | 2457 | 382 | 684 | 229 | 309 | 97 |
| 2010 | 3530 | 8196 | 2372 | 386 | 793 | 210 | 372 | 122 |
| 2011 | 3266 | 8415 | 2410 | 380 | 734 | 192 | 326 | 116 |
| 2012 | 3392 | 8744 | 2410 | 389 | 881 | 264 | 394 | 116 |
| 2013 | 3418 | 9036 | 2446 | 375 | 1120 | 288 | 525 | 175 |
| 2014 | 3255 | 10,086 | 2527 | 389 | 1548 | 395 | 689 | 285 |
| 2015 | 3033 | 10,083 | 2610 | 395 | 1897 | 505 | 874 | 381 |
| 2016 | 3201 | 10,802 | 2729 | 409 | 2457 | 567 | 1180 | 516 |
| 2017 | 3192 | 11,062 | 2845 | 428 | 1911 | 468 | 872 | 403 |
| CAGR | –4.2 | 4.6 | 1.8 | –0.4 | 16.4 | 14.3 | 14.9 | 19.7 |

[1] wild boar, moose, fallow deer, red deer, roe deer, European bison (decomposed in Table A2).

**Table A2.** Distribution of the annual ungulate–vehicle collisions (UVCs) by animal species.

| Year | Wild Boar | Moose | Fallow Deer | Red Deer | Roe Deer | European Bison |
|------|-----------|-------|-------------|----------|----------|----------------|
| 2002 | 23 | 19 | 0 | 5 | 150 | 0 |
| 2003 | 36 | 11 | 0 | 7 | 166 | 0 |
| 2004 | 42 | 14 | 0 | 6 | 263 | 0 |
| 2005 | 50 | 26 | 0 | 4 | 291 | 0 |
| 2006 | 63 | 47 | 0 | 7 | 352 | 0 |
| 2007 | 86 | 62 | 0 | 11 | 490 | 0 |
| 2008 | 100 | 60 | 0 | 6 | 485 | 0 |
| 2009 | 97 | 38 | 0 | 13 | 483 | 1 |
| 2010 | 94 | 71 | 0 | 5 | 532 | 0 |
| 2011 | 69 | 82 | 0 | 16 | 464 | 0 |
| 2012 | 89 | 85 | 0 | 10 | 583 | 0 |
| 2013 | 118 | 126 | 1 | 25 | 712 | 0 |
| 2014 | 151 | 147 | 5 | 28 | 1031 | 1 |
| 2015 | 180 | 185 | 3 | 33 | 1367 | 1 |
| 2016 | 115 | 207 | 1 | 41 | 1912 | 2 |
| 2017 | 103 | 160 | 1 | 31 | 1460 | 1 |
| CAGR | 10.5 | 15.3 | NS | 12.9 | 16.4 | NS |

**Table A3.** The number, length, and strength of ungulate–vehicle collision (UVC) hotspots on different categories of roads.

| Year | Number | | | Total Length (km) | | | Average Strength (KDE+) | | |
|------|--------|----------|----------|-------------------|----------|----------|-------------------------|----------|----------|
| | Main | National | Regional | Main | National | Regional | Main | National | Regional |
| 2002 | 2 | 2 | 0 | 279.00 | 267.00 | 0.00 | 0.49 | 0.47 | 0.00 |
| 2003 | 6 | 3 | 0 | 748.95 | 383.00 | 0.00 | 0.37 | 0.44 | 0.00 |
| 2004 | 2 | 4 | 0 | 244.67 | 492.94 | 0.00 | 0.27 | 0.40 | 0.00 |
| 2005 | 6 | 2 | 0 | 762.85 | 283.00 | 0.00 | 0.37 | 0.50 | 0.00 |
| 2006 | 8 | 10 | 0 | 972.35 | 1278.03 | 0.00 | 0.40 | 0.40 | 0.00 |
| 2007 | 10 | 16 | 0 | 1239.00 | 2105.55 | 0.00 | 0.40 | 0.42 | 0.00 |
| 2008 | 11 | 18 | 0 | 1559.51 | 2238.31 | 0.00 | 0.44 | 0.39 | 0.00 |
| 2009 | 8 | 14 | 0 | 1157.00 | 1840.59 | 0.00 | 0.50 | 0.43 | 0.00 |
| 2010 | 10 | 17 | 0 | 1347.00 | 2173.23 | 0.00 | 0.47 | 0.42 | 0.00 |
| 2011 | 6 | 10 | 3 | 734.78 | 1239.49 | 395.96 | 0.26 | 0.40 | 0.40 |
| 2012 | 8 | 11 | 2 | 1058.00 | 1437.90 | 200.34 | 0.46 | 0.40 | 0.33 |
| 2013 | 16 | 24 | 4 | 2046.77 | 2994.77 | 513.87 | 0.42 | 0.40 | 0.43 |
| 2014 | 33 | 35 | 7 | 4818.19 | 4533.86 | 916.35 | 0.42 | 0.42 | 0.33 |
| 2015 | 40 | 44 | 9 | 5715.89 | 6033.05 | 1236.00 | 0.45 | 0.45 | 0.48 |
| 2016 | 55 | 99 | 14 | 7543.99 | 13,230.43 | 1717.12 | 0.41 | 0.40 | 0.36 |
| 2017 | 40 | 64 | 18 | 5313.26 | 8573.58 | 2473.20 | 0.43 | 0.42 | 0.41 |
| CAGR | 22.1 | 26.0 | 41.4 | 21.7 | 26.0 | 96.3 | –0.9 | –0.7 | 9.9 |

**Table A4.** The number and length of new fences built on different categories of roads.

| Year | Number | | | Length of New Fences (km) | | |
|------|--------|--------|--------|--------|--------|--------|
| | **Main** | **National** | **Regional** | **Main** | **National** | **Regional** |
| 2002 | 75 | 1 | 5 | 56.13 | 0.19 | 0.34 |
| 2003 | 14 | 10 | 3 | 12.67 | 0.42 | 0.33 |
| 2004 | 7 | 6 | 5 | 0.34 | 0.29 | 0.07 |
| 2005 | 32 | 9 | 0 | 32.23 | 3.17 | 0.00 |
| 2006 | 23 | 18 | 2 | 36.74 | 16.83 | 0.51 |
| 2007 | 73 | 20 | 9 | 56.62 | 0.73 | 0.31 |
| 2008 | 100 | 35 | 4 | 184.42 | 8.15 | 0.52 |
| 2009 | 99 | 28 | 15 | 178.02 | 1.23 | 0.53 |
| 2010 | 58 | 8 | 11 | 85.81 | 0.30 | 1.59 |
| 2011 | 48 | 27 | 24 | 22.43 | 2.87 | 4.52 |
| 2012 | 22 | 9 | 20 | 11.01 | 0.80 | 0.39 |
| 2013 | 41 | 33 | 11 | 28.26 | 1.27 | 0.80 |
| 2014 | 32 | 23 | 17 | 14.35 | 1.78 | 0.68 |
| 2015 | 47 | 21 | 11 | 20.50 | 8.61 | 0.49 |
| 2016 | 3 | 8 | 15 | 3.90 | 2.01 | 0.50 |
| 2017 | 6 | 0 | 0 | 0.35 | 0.00 | 0.00 |
| CAGR | –15.5 | –14.2 | –23.0 | –28.7 | –4.2 | 2.8 |

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
