# Peer review of "Impact of Road Fencing on Ungulate–Vehicle Collisions and Hotspot Patterns"

_land, doi:10.3390/land10040338_

Round 1
Reviewer 1 Report
The manuscript assesses the impact of road fencing on ungulate-vehicle collisions along with the ungulate-vehicle collision hotspots. The subject is interesting and the expected relationship between UVC and fencing confirmed by the current research. The paper is also well written and referenced. However, for the sake of clarity, the quality of the presentation of the paper needs some improvement. I suggest to put a special emphasis on the section related to the statistical analyses. Although I do agree with the regression analyses used, some clarifications in the corresponding section are needed to avoid any misunderstanding. This is also crucial for the results and discussion sections.
Please find below some specific comments.
L31 Please give the meaning of WVC. You already defined it in L38, which is weird. Meaning of an abbreviation should always be given in the first occurrence i.e. in L31 in this paper.
L36 Using non-standard abbreviation requires giving the meaning in the first occurrence. Please give the meaning of UVC.
L43 Replace “numbers” with “number”.
L66 Meaning of AADT to be given the first time the abbreviation is used. Thereafter you can just use the abbreviation.
L79-80 This sentence needs a verb.
L107 Remove comma between “may” and “not”.
L136 We wonder initial and final value of what until we get to the Fig.5, where we understand these values are for UVC, AADT, UVC hotspots, and fence length. You may clarify it here before we get further in the paper.
L141-144 Here is how I understand what you did. The models you tested using regression are the following:
Case 1:
(Eq. 1) à UVC = AADT + Fence_length + Road_category + Error_term
(Eq. 2) à UVC_hotspots = AADT + Fence_length + Road_category + Error_term
Case 2:
An alternative understanding is to strip Road_category of Eq. 1 and Eq. 2 above and run each of the model separately for a) national, b) main and c) regional roads.
My question is the following: which of the two cases did you effectively apply? This is important for the understanding of your Tables 1 and 2 and the consecutive discussion.
L167-169 Please give the units of the variables in the Y-axis.
L171 Add space in “Figure5”.
L190-193 Is bUVC the coefficient associated with UVC variable. Please add its meaning.
L248-250 This is not explicitly shown in the Table 1 you refer to. In all cases, you need to more elaborate here.
Author Response
Dear Reviewer#1
Thank you very much for your very helpful and detailed suggestions, which helped us to clarify and improve our manuscript. We carefully discussed all your comments and did required changes to address them. Please find attached our answers to your comments. We hope that we managed to address all you concerns and could give a better flow to the manuscript text.

Reviewer 2 Report
Overview and general impressions
The topic is original, immediate and topical and with potential interest to readers, and especially policy makers and different professionals involved transport planning, road design and ecological and green infrastructure management. The research design is appropriate and the methods used are adequately described. The results are clearly presented and the discussion and the conclusions proceed the results. Illustrations and other supporting materials (tables and graphs) of high quality are provided. The structure of the paper could be improved, in terms of introducing the general reference framework of the problem examined as well as the local context. Some improvements on the methodology of the research are also possible, as well as a broader discussion on the planning and design of safety measures.
Discussion
In order to better situate the research findings, there is a need to outline the framework of the research and position it within the theory-practise dialogue on two important topics: safety on road and management of green infrastructure and ecological networks. There is also a need to briefly introduce the institutional context and the set of measures studies.
The long “term – study” with reference to data analysis is well presented, but nevertheless, the main research questions and hypothesis are missing or not well communicated. Although that formally a section dedicated to ‘materials and methods’ exists (section 2, starting on line 73), the section about the methodology of the study could be improved by clearly describing the methodological approach and the steps performed. In this line, the text between lines 63-72 can be moved to section 2. There is a need to formulate/introduce the research aim and the research questions, the potential answers of which could be associated with the subsections in section 3. It is also recommended to give a brief overview of the content of each section by summarizing the main issues addressed in the subsections – e.g. Section 2 (summarize lines 103,121,133,140), or give insight how are the results interpreted in Section 3 (summarizing lines 148,158,187).
The discussion is somehow missing the subjects (the actors) responsible for the implementation and monitoring and the mechanisms for their decisions. In that line, the whole paper will increase its polemics by outlining the gaps (if any) and the possibilities to improve the decision-making process on measures implementation, by suggestions for improving the datasets, proposing suggestions or comments on data gathering and their customization for the purposes of the road management (including planning and design).
Other remarks:
- It seems that year 2011 is a turning point – please clarify what happened then, maybe the is a legislative (or other) event that significantly impacted the trend. (It is also clearly outlined and read on Fig 4b)
- Line 86 Missing “The “before “Country” at the beginning of the sentence
- The abstract, the figures and some texts mention the time frame from 2002 to 2017. The text that introduces the country case study (line 84) uses the reference year 2012 (somewhere in the middle of the period. Why? Is there something special about this year?
Conclusion:
I came away with comments that restrict me from recommending this paper for publication as it stands. Therefore, I recommend that a minor revision be warranted. I would ask that the authors specifically address my comments.
Author Response
Dear Reviewer#2
Thank you very much for your very helpful and detailed suggestions, which helped us to clarify and improve our manuscript. We carefully discussed all your comments and did required changes to address them. Please find attached our answers to your comments. We hope that we managed to address all you concerns and could give a better flow to the manuscript text.
